# Tectoridin Stimulates the Activity of Human Dermal Papilla Cells and Promotes Hair Shaft Elongation in Mouse Vibrissae Hair Follicle Culture

**DOI:** 10.3390/molecules27020400

**Published:** 2022-01-08

**Authors:** Gary Ka-Wing Yuen, Bryan Siu-Yin Ho, Lish Sheng-Ying Lin, Tina Ting-Xia Dong, Karl Wah-Keung Tsim

**Affiliations:** 1Shenzhen Key Laboratory of Edible and Medicinal Bioresources, HKUST Shenzhen Research Institute, Shenzhen 518057, China; kwgyuen@ust.hk (G.K.-W.Y.); botina@ust.hk (T.T.-X.D.); 2Division of Life Science and Centre for Chinese Medicine, The Hong Kong University of Science and Technology, Hong Kong, China; hosyb2002@yahoo.com.hk (B.S.-Y.H.); lishlin@ust.hk (L.S.-Y.L.)

**Keywords:** *Belamcanda chinensis* (L.) DC., tectoridin, hair growth, alopecia

## Abstract

To search hair growth-promoting herbal extract, a screening platform of having HEK293T fibroblast being transfected with pTOPFLASH DNA construct was developed over a thousand of herbal extracts and phytochemicals were screened. One of the hits was ethanolic extract of Rhizoma Belamcandae, the rhizome of *Belamcanda chinensis* (L.) DC. Tectoridin, an isoflavone from Rhizoma Belamcandae, was shown to be responsible for this activation of promoter construct, inducing the transcription of pTOPFLASH in the transfected fibroblasts in a dose-dependent manner. The blockage by DKK-1 suggested the action of tectoridin could be mediated by the Wnt receptor. The hair growth-promoting effects of tectoridin were illustrated in human follicular dermal papilla cells and mouse vibrissae organ cultures. In tectoridin-treated dermal papilla cultures, an activation of Wnt signaling was demonstrated by various indicative markers, including TCF/LEF1 transcriptional activity, nuclear translocation of β-catenin, expressions level of mRNAs encoding axin-related protein, (AXIN2), β-catenin, lymphoid enhancer-binding factor-1 (LEF-1), insulin-like growth factor 1 (IGF-1) and alkaline phosphatase (ALP). In addition, an increase of hair shaft elongation was observed in cultured mouse vibrissae upon the treatment of tectoridin. Tectoridin, as well as the herbal extract of Rhizoma Belamcandae, possesses hair promoting activity, which deserves further development.

## 1. Introduction

Hair growth is a dynamic process being controlled by hair follicle in a cyclic manner. The growth of hair has several important functions in social interaction, psychological well-being, and regulating body temperature [1]. Hair follicle undergoes the cycles of growth and regression throughout the lifetime. The hair follicle cycle is mainly classified into three phases: (i) the active hair growth phase (anagen); (ii) the regression phase (catagen); and (iii) the resting phase (telogen) [2]. The control of hair growth cycle relies on a delicate balance between growth promoting and inhibiting signals. The Wnt/β-catenin signaling is known to play a role in the developmental processes of hair [3], and the activation of Wnt/β-catenin signaling triggers the growth of hair follicle, as well as to stimulate the expressions of genes at the anagen phase of hair cycle [4]. The hair growth-promoting signals have been identified, including the signaling of wingless (Wnt), fibroblast growth factor (FGF), vascular endothelial growth factor (VEGF), insulin-like growth factor 1 (IGF-1). In contrast, transforming growth factor β (TGF-β) and bone morphogenetic protein (BMP) are known to inhibit hair growth or to drive hair follicle into regression [5]. Dermal papilla cell (DPC), a cluster of specialized mesenchymal cells located at the base of hair follicle, plays an important role in regulating hair growth [6], which secretes different paracrine and autocrine factors to regulate the growth cycle of hair follicle, as well as various intracellular activities inside the follicles. DPC provides different signals of Wnt, R-spondin, FGF, and Noggin in activating the follicular bulge stem cell, and thereafter which initiate the anagen phase [7].

Alopecia is the condition of progressive loss of terminal hair on the scalp or anywhere on the body [8]. Alopecia can be mainly divided into two categories, scarring and non-scarring [9]. Scarring alopecia refers to permanent hair loss, and the hair follicles are irreversibly destroyed, e.g., lichen planopilaris and folliculitis decalvans. Non-scarring alopecia refers to hair loss due to the hair cycle being disrupted, as in cases of alopecia areata, telogen effluvium and androgenic alopecia. In non-scarring alopecia, hair follicles are preserved, and which can be reactivated for hair regrowth [10]. During embryonic and adult stages, the Wnt/β-catenin signaling plays an important role in hair morphogenesis and cycling [3,11]. Wnt3a and Wnt7b containing macrophage-extracellular vesicles can increase the proliferation of DPC and enhance hair follicle growth in male mice in vivo [12,13]. Thus, activators of Wnt/β-catenin signaling could be considered for the treatment of hair loss. Several herbal extracts and phytochemicals have been shown to promote hair growth through Wnt/β-catenin signaling. For example, baicalin was shown to promote hair growth in mouse and DPC via Wnt/β-catenin signaling [14]. A Wnt/β-catenin activator (SM04554) has entered the clinical trial in treating androgenetic alopecia [15].

Tectoridin is a major isoflavone from the rhizome of *Belamcanda chinensis* (L.) DC., a Chinese medicinal herb of Iridaceae family named as Rhizoma Belamcandae. In Chinese medicine, Rhizoma Belamcandae is commonly used to treat asthma, tonsillitis, and inflammation [16]. Tectoridin has been shown to have estrogenic, anti-oxidative and hepatoprotective effects [17,18], as well as its functions in osteoblastogenesis and osteoclastogenesis [19]. Here, we reported a novel capability of tectoridin in promoting hair growth by activating the Wnt signaling.

## 2. Results

### 2.1. Extracts of Rhizoma Belamcandae and Tectoridin Activate Wnt/β-Catenin Signalling

The pTOPFLASH DNA construct containing the TCF-binding consensus sequence for β-catenin TCF complex was used to quantify the activation of Wnt/β-catenin signaling by measuring the driven luciferase activity [20]. In pTOPFLASH-transfected HEK293T cells, the herbal extracts and phytochemicals were applied for 24 h for activation of luciferase activity. Over 1000 herbal extracts and phytochemicals were screened via the platform. Among over 100 positive hits, the extracts of Rhizoma Belamcandae showed robust activation on the assay of pTOPFLASH DNA construct, and thus which was chosen for further analysis.

Three herbal extracts, i.e., water, 50% ethanol, and 70% ethanol, from Rhizoma Belamcandae were tested in pTOPFLASH-transfected HEK293T cells. To control the herbal extracts, HPLC fingerprinting was employed, showing the chemical profile and amount of tectoridin (Figure 1A). The linear regression equation was used to quantify the amount of tectoridin in the extracts. The linear regression data had a good linear relationship with the correlation coefficients (*r^2^*) of tectoridin which was 0.9984. The linear calibration equation was Y = 42.26X + 15.413, where Y was the peak area, and X was the concentration of tectoridin. The content of tectoridin in water extract, 50% ethanol extract and 70% ethanol extract of Rhizoma Belamcandae were determined as 0.63 ± 0.08, 1.71 ± 0.13, 9.73 ± 1.34 mg/g, respectively, where *n* = 4.

In dose-dependent manner, the herbal extracts induced the luciferase activity in cultured HEK293T cells having transfection of pTOPFLASH DNA construct (Figure 1B). The luciferase activity, induced by the 70% ethanol extract, showed the highest induction among the others, having a maximal activation by ~3.5 folds at 150 μg/mL of the herbal extract. The activation of luciferase was in an order of 70% ethanol extract > 50% ethanol extract > water extract.

Rhizoma Belamcandae contains a number of phytochemicals, e.g., isoflavones, xanthone glycosides, stilbenes, simple phenols, and quinones [16,17,18]. Having AXIN2 as a common drug target of Wnt/β-catenin signaling for molecular docking [21], we tested the possible interaction of phytochemicals, i.e., tectoridin, resveratrol, mangiferin, tectorigenin, irigenin, 7-*O*-methylmangiferin, and irisflorentin; the phytochemicals were identified in Rhizoma Belamcandae [22]. Tectoridin was predicted as the lowest estimated free energy for the binding (−12.3 KJ/mol), as compared to others (Appendix A).

These phytochemicals of Rhizoma Belamcandae, i.e., tectoridin, resveratrol, mangiferin, tectorigenin, irigenin, 7-*O*-methylmangiferin, and irisflorentin, were applied to pTOPFLASH-transfected HEK293T cells (Appendix A). Resveratrol, irigenin, tectorigenin, and tectoridin showed an increase of the luciferase activities. However, tectoridin was the highest content of flavone in the ethanol extracts of Rhizoma Belamcandae, while resveratrol, irigenin, and tectorigenin were close to minimal level (Figure 1A). Besides, the content of tectoridin, Rhizoma Belamcandae was at least few folds higher than the others [22].

Here, we hypothesized that tectoridin could be the active chemical of Rhizoma Belamcandae in triggering the DNA construct. In pTOPFLASH-transfected HEK293T cells, tectoridin was applied in different doses (Figure 2A). Up to 200 µM of tectoridin, the cell viability of cultured HEK293T cells did not change (Appendix A). Applied tectoridin induced the luciferase activity in a dose-dependent manner at a dose below 20 µM: the maximal induction was at ~80% increase, as compared to the background (Figure 2A). DKK-1, an inhibitor of the Wnt/β-catenin signaling by binding to LRP5/6 directly to prevent binding of Wnt to its receptors, was used here to test the specificity of tectoridin to the Wnt receptor. The transfected cells were pre-treated with DKK-1 for 1 h before the application of Wnt3a or tectoridin. The luciferase activity, induced by Wnt3a or tectoridin, was fully inhibited after the pre-treatment with DKK-1 (Figure 2B). The result indicated that tectoridin was acting upstream of Wnt/β-catenin signaling or directly binding to the Wnt receptor.

Accumulation of β-catenin in nucleus is a key step in the Wnt/β-catenin signaling pathway. After translocation to the nucleus, β-catenin binds to T-cell factor (TCF) and activates the transcription of downstream target genes. In transiently transfected HEK293T cells having constitutive expression of β-catenin-GFP, the effect of tectoridin on β-catenin subcellular trans-localization was determined (Figure 2C). Valproic acid (VPA) is known to be an inhibitor of GSK-3β [23], serving as a positive control. The localization of β-catenin-GFP was more nuclear-restricted in transfected cells being treated with VPA, or tectoridin, after 24 h, as compared to the control group, suggesting the translocation of β-catenin under the drug treatment.

### 2.2. Tectoridin Induces Hair Growth

DPC is the most common cell model in testing the signaling of hair growth. In cultured DPC, the role of tectoridin was tested in inducing specific gene expressions related to hair growth. The mRNA levels of genes downstream in Wnt signaling pathway, i.e., axin-related protein, (AXIN2), β-catenin, lymphoid enhancer-binding factor-1 (LEF-1), insulin-like growth factor 1 (IGF-1), and alkaline phosphatase (ALP), were revealed by RT-PCR after the treatment of tectoridin for 48 h in the cultures. The results indicated that the levels of mRNAs encoding AXIN2, LEF-1, and β-catenin were significantly increased in a dose-dependent manner upon treatments with tectoridin (Figure 3). In parallel, the growth biomarker of DPC, ALP, was also increased by tectoridin significantly. Moreover, the expression level of IGF-1, a potent growth factor in promoting hair growth, was increased by over two-folds upon the treatment (Figure 3). Wnt3a served as a positive control here in activating the gene expression.

Vibrissae hair follicles from mouse is a common ex vivo model in testing hair growth. To evaluate the hair growth promoting property of tectoridin, the isolated vibrissae hair follicles were treated with different doses of tectoridin for 72 h (Figure 4). The length of hair was measured during the growth, and which was significantly elongated in a dose-dependent manner, as compared to control follicle. Under the treatment of 50 or 100 µM tectoridin for 3 days, the hair could grow by over 60% (Figure 4): the result was in parallel to that in cell cultures. WAY 316606, an inhibitor of secreted frizzled-related protein-1 (sFRP-1) preventing sFRP-1 interacting with Wnt, served as a positive control showing an induction of hair length by ~25%.

## 3. Discussion

Different forms of hair loss including androgenetic alopecia, senescent alopecia, and telogen effluvium are common in the population over 50 years of age. Currently approved medications, such as Minoxidil and Finasteride, exhibited limited efficacy and unwanted side effects, prompting the search for alternative treatments for hair loss. The hair follicle undergoes cycles of growth, regression, and resting phase throughout life, and the transition of different phases is tightly regulated by DPC, acting as the signaling center of hair growth. Besides, Wnt signaling is well-known to play such a role in activating hair growth [20]. The activation of Wnt signaling is initiated by binding of Wnt ligands, including Wnt3a, to Frizzled and LRP5/6 co-receptors to stabilize the cytoplasmic β-catenin by inducing GSK3β phosphorylation. The translocation of β-catenin into the nucleus activates TCF/LEF and downstream target genes. By drug screening platform, we have identified a possible Wnt activator, tectoridin. This notion is supported by its effects in cultures: (i) inducing β-catenin translocation to nucleus; (ii) activating transcriptions of Wnt signaling-mediated genes and blocked by Wnt receptor inhibitor; and (iii) inducing outgrowth of hair.

This is the first time to identify the hair growth-inducing property of Rhizoma Belamcandae. This medicinal herb has been used as a common traditional Chinese medicine for years. According to historical records, the herb is tailored for various medications, including antipyretic, anti-inflammatory, antidiabetic, anti-angiogenic, and anti-tumor properties [24]. The major phytochemicals in Rhizoma Belamcandae are polyphenols, e.g., isoflavones, xanthone glycosides, stilbenes, simple phenols, and quinones. Indeed, tectoridin is the major isoflavone in Rhizoma Belamcandae, accounting for 4 to 2570 mg/kg of crude herb [24]. Tectoridin has been proposed to account for the activities of Rhizoma Belamcandae, which has exhibited multiple properties, including anti-microbial and anti-fungal activity, anti-inflammatory via inhibition of COX2 activity [25], as well as its estrogenic property [26]. The possible role of tectoridin in alleviating multi-factorial pathogenesis of androgenetic alopecia has been reported. In patients suffering from androgenetic alopecia, inflammation is commonly reported [27]. Administration of COX2 inhibitor has been employed to restore hair growth [28]. Indeed, tectoridin has been demonstrated to possess anti-inflammatory properties, including inhibition of COX2 [25], which therefore could account partly for the hair growth stimulation, besides the action on Wnt signaling. Tectoridin is considered as phytoestrogen having known estrogenic effect: this property may be beneficial in reversing hair loss in females suffering from androgenetic alopecia [29]. Thus, Rhizoma Belamcandae and tectoridin could have a great potential to be developed as hair growth products for androgenetic alopecia, which have the advantage of not having the side effects as the current drugs.

The activation of Wnt signaling by another Chinese herb, roots of *Pueraria thomsonii Benth.*, has been reported in cultured HS68 cell, a foreskin-derived skin fibroblast cell line [30]. Tectoridin, also a significant isoflavone in the roots of *P. thomsonii*, has been proposed to account for this Wnt activation, and which can suppress DKK-1, an inhibitor of Wnt signaling, with regard to the activation of the Wnt signaling. In methanol extract, the tectoridin content in the root of *P. thomsonii* is similar to that of Rhizoma Belamcandae, i.e., 2.97 mg/g and 2.8 mg/g, respectively [30,31]. In our result, the content of tectoridin in Rhizoma Belamcandae can reach 9.7 mg/g using 70% ethanol extraction. Thus, application of extracts deriving from Rhizoma Belamcandae, and/or roots of *P. thomsonii*, in treating androgenetic alopecia could be considered.

## 4. Materials and Methods

### 4.1. Materials

Cell culture media and supplements were purchased from Thermo Fisher Scientific (Waltham, MA, USA), except for those that were specifically indicated. Tectoridin was purchased from Chengdu Herbpurify Ltd. (Chengdu, China) with a purity ≥98%. Tectoridin was dissolved in dimethyl sulfoxide (DMSO) to give a stock solution of 50 mM. Wnt3a and DKK-1 were purchased from R&D Systems (Minneapolis, MN, USA). 3-(4,5-Dimethylthiazol-2-yl)-2,5-diphenyl tetrazolium bromide (MTT), and valproic acid (VPA) were purchased from Sigma-Aldrich (St. Louis, MO, USA). WAY 316,606 was purchased from ApexBio (Houston, TX, USA).

### 4.2. Molecular Docking

Chemical structures of molecules were downloaded from Pubchem (https://pubchem.ncbi.nlm.nih.gov/, accessed on 2 November 2021); while the structure of AXIN2 was downloaded from Protein Data Bank (PDB, https://www.rcsb.org/, accessed on 1 November 2021). Virtual screening was performed on SEESAR software (www.biosolveit.de/LeadIT, accessed on 18 October 2021) following the procedures as below: (i) the binding site was defined according to the residues composing the identified druggable pocket. Ligand states, including protonation and tautomeric forms, were automatically assessed in the model using ProToss method, which subsequently generated the most accessible hydrogen positions based on an optimal hydrogen bonding network; (ii) the docking simulation was performed on “Compute LeadIT Docking” mode by using FlexX algorithm. Ten binding conformations for each ligand were generated and (iii) the “assess affinity with HYDE (hydrogen bond and dehydration) in SEESAR” node produced refined binding free energy (i.e., ∆G) and estimated HYDE affinity (KiHYDE) for each ligand pose using HYDE rescoring function.

### 4.3. Raw Material and HPLC Condition

The raw material of Rhizoma Belamcandae (rhizome of *B. chinensis*) was obtained from Guangdong province in 2019 and authenticated by Dr. Tina Dong, one of the authors. The authentication of the herbs was according to Hong Kong Materia Medica Standards. The voucher specimen was deposited at Centre for Chinese medicine at the university. The raw material of Rhizoma Belamcandae was weighed and sonicated with water, 50% ethanol or 70% ethanol for 30 min, twice: the volume was 15 times and 10 times, respectively. The herbal extract was dried by a Labconco FreeZone Freeze Dry System. The dried extract was then dissolved in its extraction solution to a final concentration of 100 mg/mL. HPLC-UV chromatographic separation of extracts deriving from Rhizoma Belamcandae was performed on an Agilent HPLC 1200 series system (Agilent, Waldbronn, Germany) equipped with a degasser, a binary pump, an autosampler, a thermos-stated column compartment, and a diode array detector. The herbal extracts were filtered by a 0.22-µm filter before separated by an ACE 5 C18 HPLC Column (particle size 5 μm, 4.60 mm × 250 mm). The mobile phase was composed of acetonitrile (A) and 0.05% phosphoric acid in water (B). According to the following gradient program: 0–15 min, linear gradient 18–20% (A); 15–25 min, linear gradient 20–33% (A); 25–45 min, linear gradient 33–40% (A); 45–60 min, linear gradient 40–53% (A); and 60–65 min, isocratic gradient 53% (A). A pre-balance period of 5 min was used between each run. The flow rate was set at 1 mL/min; the column temperature was 25 °C; and the injection volume was 10 µL. A DAD detector at an absorbance of 266 nm was used.

### 4.4. Cell Culture

Immortalized DPCs were obtained from Applied Biological Materials (Richmond, BC, Canada). DPC and human embryonic kidney (HEK 293T) cells were cultured in Dulbecco’s modified Eagle medium supplemented with 10% fetal bovine serum (FBS) and 1% (*v*/*v*) penicillin/streptomycin (stock as 10,000 U and 10,000 mg/mL) in 5% CO_2_ at 37 °C. All culture reagents were purchased from Thermo Fisher Scientific.

### 4.5. Animals

Animals were obtained from Animal and Plant Care Facility of The Hong Kong University of Science and Technology (HKUST) and tested according to the guidelines of Department of Health, The Government of Hong Kong SAR. The experimental procedures were approved by Animal Ethics Committee at the university (reference no.: [20,21,22,23,24,25] in DH/HT&A/8/2/2 Pt.1.).

### 4.6. Cell Proliferation Assay

Cell proliferation was assessed by MTT (3-(4,5-dimethylthiazolyl-2)-2,5-diphenyltetrazolium bromide) assay. DPC and HEK293T cells were seeded in 96-well plates at a density of 5000 cells per well for 24 h, and then were treated with tectoridin, or extracts from Rhizoma Belamcandae, for 24 h. Afterward, the medium was removed, and 100 μL of MTT (0.5 mg/mL in complete growth medium) solution was added to each well. Then, cells were incubated at 37 °C for 1 h. Following the incubation, the medium was replaced with 100 μL of DMSO in each well and shaken for 15 min. Absorbance was measured using a Multiskan FC Microplate Photometer (Thermo Fisher Scientific) at a wavelength of 570 nm.

### 4.7. DNA Transfection and Luciferase Reporter Assay

The plasmid having TCF/LEF-firefly luciferase reporter (pTOPFLASH) with two repeats, each containing three copies of the TCF-binding site upstream of thymidine kinase minimal promoter, was purchased from Upstate Biotechnology (Lake Placid, NY, USA). Green fluorescent protein (GFP)-tagged β-catenin construct was a kind gift from Dr. Henderson (University of Sydney, Australia). The reporter construct was then transfected using jetPRIME kit Transfection Reagent (Polyplus transfection, New York, NY, USA) according to the instruction stated on the kit. For the luciferase reporter assay, luciferase assay was performed using PierceTM Firefly Luciferase Glow Assay Kit (Thermo Fisher Scientific Inc., Waltham, MA, USA) according to the instruction stated on the kit. The luminescent reaction was quantified in a GloMax^®^ 96 Microplate Luminometer (Thermo Fisher Scientific Inc., Waltham, MA, USA). All values were presented as firefly luciferase activity normalized to the total protein of the samples.

### 4.8. Fluorescence Microscopy

To observe the localization of GFP-tagged β-catenin, the cells were grown on 10-mm glass coverslips, transfected and incubated as indicated above, then washed twice in PBS, immediately fixed with 4% (*v*/*v*) paraformaldehyde in PBS for 15 min at room temperature, and subsequently permeabilized with 0.2% (*v*/*v*) Triton X-100 in PBS for 10 min at room temperature. Samples were mounted with ProLong Gold Antifade Mountant with or without DAPI (Thermo Fisher Scientific). Samples were then examined by Zeiss Celldiscoverer 7 automated microscope (Zeiss, Oberkochen, Germany).

### 4.9. RT-PCR

Total RNA was extracted from cultured DPCs using RNAzol™ Reagent (Sigma-Aldrich, St. Louis, MO, USA), and 3 μg of RNA was reverse transcribed using PrimeScript™ RT reagent kit (Takara), according to manufacturer’s instruction. Template cDNAs were subjected to RT-PCR using the following specific primers: ALP(5′-AGC ACT CCC ACT TCA TCT GG-3′ and 5′-TGT CTT CCG AGG AGG TCA AG-3′); LEF-1 (5′-TTC CTT GGT GAA CGA GTC TG-3′ and 5′-GTG TTC TCT GGC CTT GTC GT -3′); AXIN2 (5′-GGG AGA AAT GCG TGG ATA C-3′ and 5′-CTG CTT GGA GAC AAT GCT GT-3′); IGF-1 (5′-TCA GAA GGG TAG CCC CTA GCT-3′ and 5′-TCA AGC CTG GGT ACT TTT AAC CA-3′); β-catenin (5′-CCC ACT AAT GTC CAG CGT TT-3′ and 5′-AAC CAA GCA TTT TCA CCA GG-3′); glyceraldehyde 3-phosphate dehydrogenase (GAPDH) (5′-ACC TGA CCT GCC GTC TAG AA-3′ and 5′-TCC ACC ACC CTG TTG CTG TA-3′). Here, GAPDH, a housekeeping gene, was used as an internal control. RT-PCR was performed in LightCycler 480 (Roche Molecular Biochemical, Indianapolis, IN, USA) using KAPA SYBR FAST qPCR kits in accordance with the manufacturer’s instruction. The 2^ΔΔC^_T_ method was used to calculate the relative expression levels.

### 4.10. Vibrissae Hair Follicle Culture

Mouse vibrissae hair follicles in anagen phase were carefully isolated from the upper lip pad of 4-week-old C57BL/6 male mice. The hair follicles were cultured in Williams E medium (Sigma-Aldrich) supplemented with 1% (*v*/*v*) penicillin/streptomycin solution in 5% CO_2_ at 37 °C. Five hair follicles were isolated for each group. Vibrissae hair follicles were incubated in various concentrations of tectoridin or 2 μM WAY-316606 (Tocris Bioscience, Ellisville, MO, USA) for 3 days, and increase in the length of hair was measured from days 0 to 3.

### 4.11. Statistical Analysis and Other Assays

Protein concentrations were measured by a kit from Bio-Rad (Richmond, CA, USA). Each result was presented as the mean ± SEM, calculated from 3 to 5 independent samples, with triplicated. Comparisons of the mean for untreated control cells and treated cells were analyzed using one-way analysis of variance (ANOVA) and Student’s t test. Significant values were represented as *, *p* < 0.05, **, *p* < 0.01, ***, *p* <0.001.

## 5. Conclusions

Tectoridin and extracts of Rhizoma Belamcandae were shown to activate Wnt/β-catenin signaling greatly in the assay of pTOPFLASH-transfected cells. Tectoridin was shown to have the capability of activating Wnt/β-catenin signaling in human DPC and promoting hair growth in vibrissae hair follicle. Our working hypothesis is the possible intervening of AXIN’s role in triggering the Wnt/β-catenin signaling by tectoridin (Figure 5). The results suggest a new possible topical medical application of using tectoridin, or extracts of Rhizoma Belamcandae, in treating alopecia. Further work must be carried out in characterizing the multiple pathways of tectoridin/Rhizoma Belamcandae extract acting on the hair follicle and the efficacy in treating androgenetic alopecia.

## Figures and Tables

**Figure 1 molecules-27-00400-f001:**
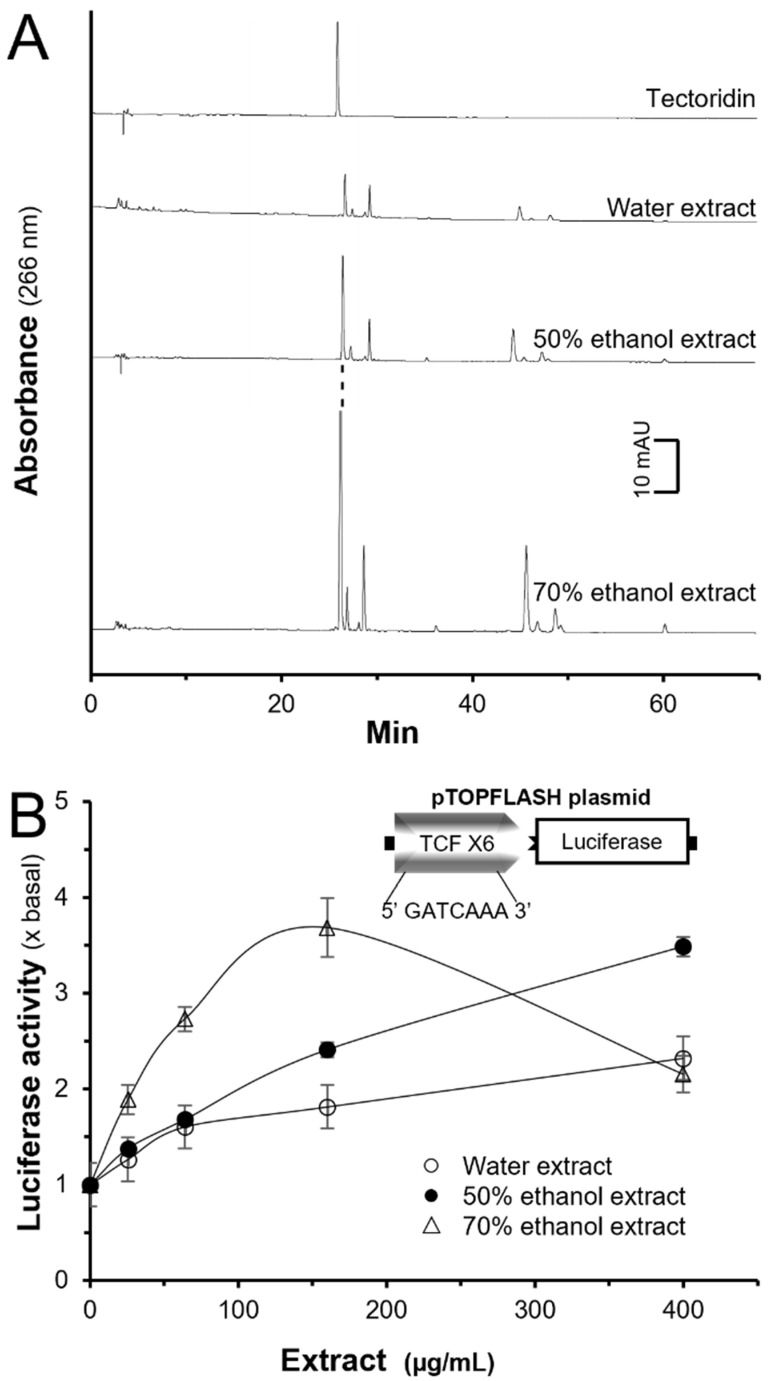
Chemical fingerprints of extracts deriving from Rhizoma Belamcandae and its role in activating Wnt/β-catenin signaling. (**A**): Different extracts of Rhizoma Belamcandae (10 mg/mL, injection volume = 10 μL) were subjected to HPLC-DAD analysis. The peaks were revealed at 266 nm. Tectoridin at 20 mg/L was used as the chemical marker here. Representative figures are shown, *n* = 4. (**B**): HEK293T cells were transfected with pTOPFLASH DNA construct (insert) for 4 h. The cultures were treated with different extracts for another 24 h. The cell lysate was subjected for luciferase assays. Data are normalized and expressed as the fold (x basal) of control (control group was treated with 0.02% DMSO), in mean ± SEM, *n* = 4, each with duplicate.

**Figure 2 molecules-27-00400-f002:**
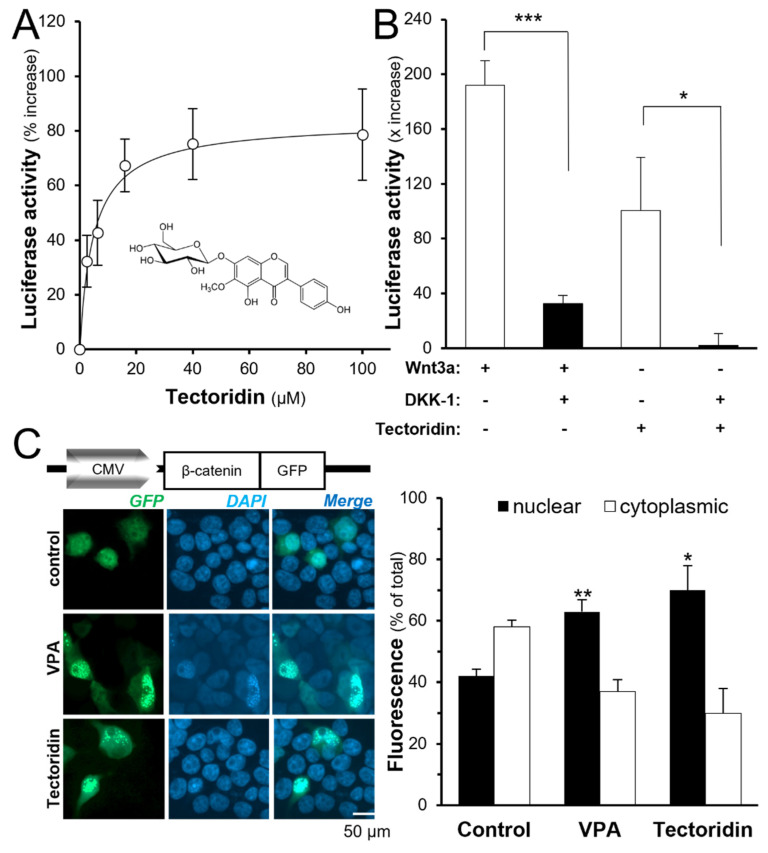
Tectoridin activates Wnt/β-catenin signaling. (**A**) HEK293T cells were transfected with pTOPFLASH DNA construct for 4 h. The cultures were treated with different doses of tectoridin for another 24 h. The cell lysate was used for luciferase assays. (**B**) HEK293T cells were transfected with as in (**A**). Then, the cultures were pre-treated with or without DKK-1 (200 ng/mL) for 1 h before the treatment of tectoridin (20 μM), or Wnt3a (200 ng/mL) for 24 h. (**C**) HEK293T cells were transfected with a DNA construct containing β-catenin tagged with green fluorescent protein (GFP; see insert), followed by treatment of tectoridin (50 µM) or valproic acid (VPA; an activator of Wnt receptor; 5 mM) for 24 h. Data are normalized and expressed as the % of increase, or total, in comparison to control (control group was treated with 0.02% DMSO), in mean ± SEM, *n* = 4, each with duplicate samples. * *p* < 0.05, ** *p* < 0.01, *** *p* < 0.001.

**Figure 3 molecules-27-00400-f003:**
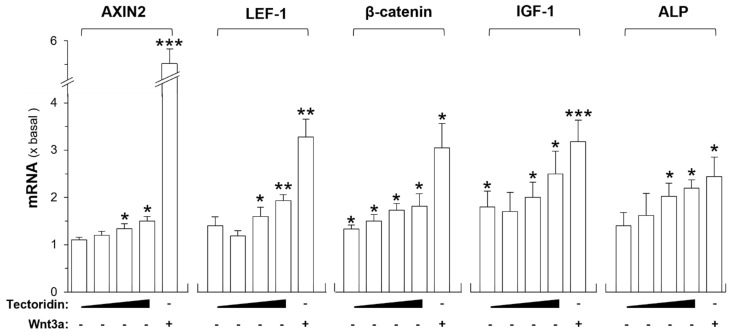
Tectoridin increases expression of downstream target genes of Wnt/β-catenin signaling. In cultured DPC, different doses of tectoridin (3, 10, 20 and 50 µM, as indicated), or Wnt3a (200 ng/mL), were applied for 48 h. The isolated total RNA from the culture was subjected to RT-PCR analyses of AXIN2, LEF-1, β-catenin, IGF-1, and ALP. Data are normalized and expressed as the fold (x basal) of control (control group was treated with 0.02% DMSO), in mean ± SEM, *n* = 5, each with duplicate samples. * *p* < 0.05, ** *p* < 0.01, *** *p* < 0.001.

**Figure 4 molecules-27-00400-f004:**
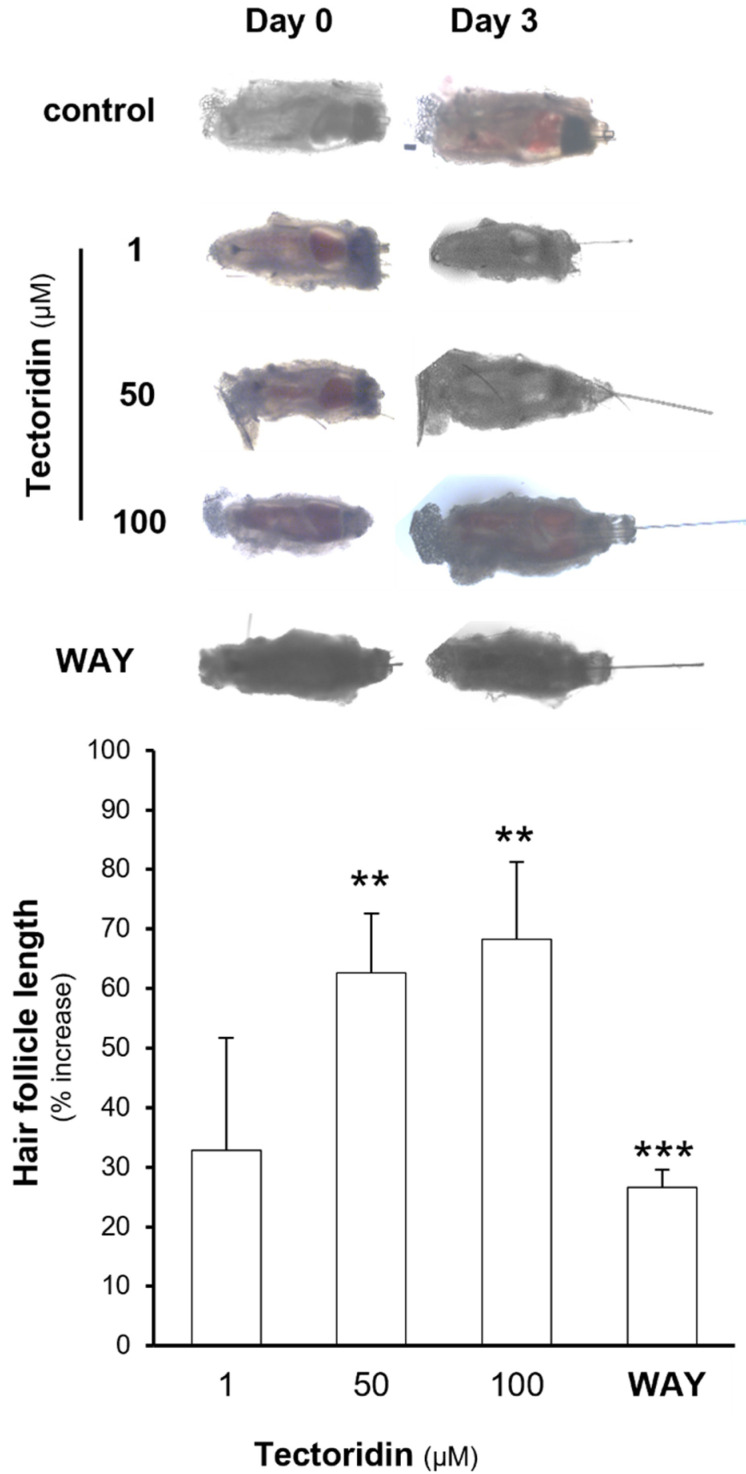
Tectoridin promotes hair shaft elongation. Individual anagen vibrissae hair follicles were isolated from the upper lip pad of 4-week-old C57BL/6 male mice and cultured in William’s medium E in 5% CO_2_ at 37 °C. The cultures were treated with different doses of tectoridin, or WAY316606 (2 μM), as indicated for 3 days. Hair shaft elongation was measured from bottom of hair follicle to the epidermis (upper panel). The measurement of hair length from the hair follicles was performed (low panel). Data are normalized and expressed as the % of increase in comparison to control (control group was treated with 0.02% DMSO), in mean ± SEM, *n* = 4. ** *p* < 0.01, *** *p* < 0.001.

**Figure 5 molecules-27-00400-f005:**
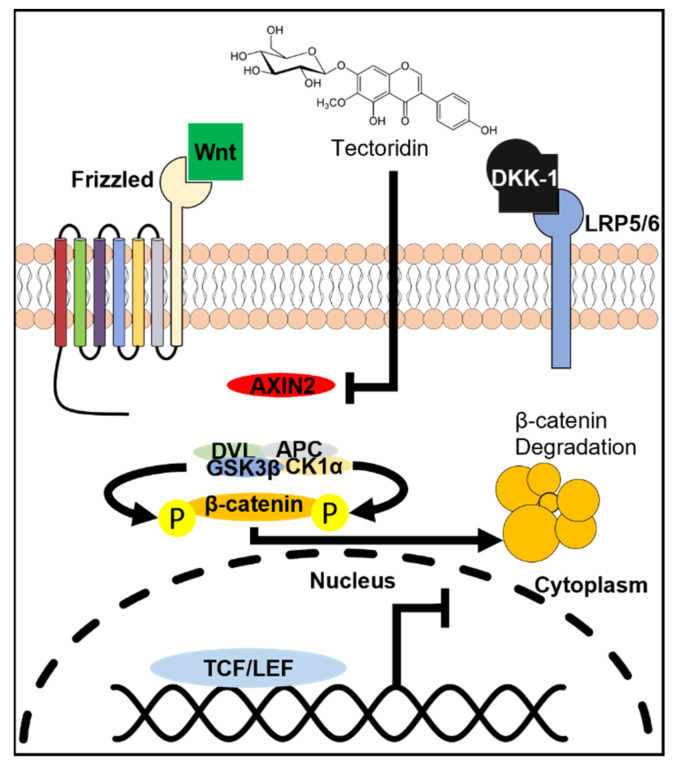
Proposed action of tectoridin in Wnt/β -catenin signaling.

## Data Availability

The data presented in this study are available upon request from the corresponding author.

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
