# Peer review of "Tectoridin Stimulates the Activity of Human Dermal Papilla Cells and Promotes Hair Shaft Elongation in Mouse Vibrissae Hair Follicle Culture"

_molecules, 2022, doi:10.3390/molecules27020400_

Round 1

Reviewer 1 Report

After the minor modification the paper can be accepted.

comments

  1. Avoid the Typo error. Ex. Line 263: CO2 has to change into CO2
  2. The picture quality is poor, the clarity or the resolution should be increased.
  3. The authors can also include the diagrammatic explanation of the mechanism of Tectoridin action on promoting hair shaft elongation can provide the best understanding for the reader.
  4. Line 303: 3***g means?
  5. The author should include a conclusion at the end of the manuscript.

Author Response

Dear Reviewer 1,

I am writing to re-submit a manuscript entitled ‘Tectoridin stimulates the activity of human dermal papilla cells and promotes hair shaft elongation in mouse vibrissae hair follicle culture’ by Yuen et al. for consideration for publication as original research in molecules - Special Issue "The Pharmacological Properties and Therapeutic Use of Natural Products". We have revised the manuscript to fill-ful the comments raised by you. Please kindly see the attachment for the manuscript.

Response to Reviewer 1 Comments

Point 1: Avoid the Typo error. Ex. Line 263: CO2 has to change into CO2

Response 1: We have revised the typos throughout the text.

Point 2: The picture quality is poor, the clarity or the resolution should be increased.

Response 2: In accordance to this comment, we have provided a better photo for illustration.

Point 3: The authors can also include the diagrammatic explanation of the mechanism of Tectoridin action on promoting hair shaft elongation can provide the best understanding for the reader.

Response 3: In accordance to this recommendation, we have included a new Figure 5 to show the proposed action of tectoridin.

Point 4: Line 303: 3***g means?

Response 4: We have revised the typo, which should be “3 μg of RNA was reverse transcribed…”

Point 5: The author should include a conclusion at the end of the manuscript.

Response 5: The conclusion has been included.

Thanks for your time and consideration. Best wishes.

Yours Sincerely,
Karl W.K. Tsim, Ph.D.
Chair Professor
Division of Life Science
The Hong Kong University of Science and Technology
Clear Water Bay Road
Hong Kong, China

Reviewer 2 Report

Dear Authors!

My comments are in attached  file.

Author Response

6 December 2021

Dear Reviewer 2,

I am writing to re-submit a manuscript entitled ‘Tectoridin stimulates the activity of human dermal papilla cells and promotes hair shaft elongation in mouse vibrissae hair follicle culture’ by Yuen et al. for consideration for publication as original research in molecules - Special Issue "The Pharmacological Properties and Therapeutic Use of Natural Products". We have revised the manuscript to fill-ful the comments raised by you. Please kindly see the attachment of the manuscript.

Response to Reviewer 2 Comments

Reviewer #2 General assessment and major comments:

Point 1: The authors state that tectoridin is the main biologically active compound with the described effects in the extract. However, a detailed analysis of other components, including those of polyphenolic nature, which, even in minor concentrations, are capable of exerting effects on living systems, has not been given. Fig 1 A demonstrates that the extract contains additional components that the authors did not mention. Although tectoridin itself has some effects, this does not prove that it is the main active ingredient of the extract. In my opinion, it is impossible to draw the conclusions without appropriate controls for other possible active components of the extracts.

Response 1: The choice of tectoridin was based on ours preliminary screening by molecular docking targeting Wnt/β-catenin signalling with the phytochemicals of Rhizoma Belamcandae. Few major phytochemicals, e.g. tectoridin, resveratrol, mangiferin, tectorigenin, irigenin, 7-O-methylmangiferin and irisflorentin, from Rhizoma Belamcandae have been subjected to molecular docking analysis in predicting the binding affinities with AXIN, a common activator together with LRP6 for Wnt/β-catenin signalling. Tectordin was predicted having the lowest estimated free energy for binding (-12.3 KJ/mol) among all major phytochemicals in Rhizoma Belamcandae. (Supplementary Figure S1). Besides, the content of tectoridin was found to be the highest among the different phenolic compounds in the extract of Rhizoma Belamcandae (at a concentration of about 2.32-19.08 mg/g). The additional description regarding tectoridin has been included in line 98 to 105.

Point 2: The authors do not explain why such an object of research was chosen as a hair growth model. And how can the results be applied to the practical use in the correction of alopecia.

Response 2: Immortalized dermal papilla cell was used as the hair growth model in demonstrating tectoridin increases expression of downstream target genes of Wnt/β-catenin signalling in Figure 3. Ex vivo mouse vibrissae hair follicle was another hair growth model being used in demonstrating tectoridin in hair growth (Figure 4). Both models are routinely used for hair growth experiments, and we just followed the common practise here. In view of this concern, we have included a clarification in the experimental part.

Reviewer #2 Minor Comments:

Point 1:

Results

2.1. Extracts of Rhizoma Belamcandae and tectoridin activate Wnt/β-catenin signalling

The first paragraph has nothing to do with the stated results and should be referred to the Discussion.

Response 1: Thanks for the suggestion. We have revised the first paragraph, and part of that has been included in the discussion.

Point 2: Fig 4 – the upper panel contains uninformative pictures. You should either improve the quality of the photos, or exclude them.

Response 2: The photo quality has been revised.

Point 3: 4.2. Raw material and HPLC condition Line 241 The authentication of the herbs was according to Hong Kong Materia Medica Standards. A reference to these standards or a short description is required.

Response 3: We have authenticated the herbs according to the description of Hong Kong Materia Medica Standards in line 273.

Point 4:

4.7. Fluorescence Microscopy; Line 299 The question arises why such a complex procedure of cell fixation was used, considering that DAPI staining can be successfully performed even in living cells.

Response 4: The function of 4% (v/v) paraformaldehyde was to preserve the specimen from decay. After that, the cell was permeabilized with 0.2% (v/v) Triton X-100 in PBS for 10 min. ProLong Gold Antifade Mountant, a product from Thermo Fisher Scientific containing DAPI, was added at the end. The complex procedure was to ensure the better quality of micrographs.

Point 5: Concluding remarks are absent. It would be useful to add this section as the Discussion section contains no formalized concluding remarks.

Response 5: A conclusion has been included.

Thanks for your time and consideration. Best wishes.

Yours Sincerely,
Karl W.K. Tsim, Ph.D.
Chair Professor
Division of Life Science
The Hong Kong University of Science and Technology
Clear Water Bay Road
Hong Kong, China
E-mail: [email protected]

Round 2

Reviewer 2 Report

Dear Authors!

I thank you for your comments. But I still have some remarks (in the attached file).
